# Peer review of "Snowmass2021 Cosmic Frontier White Paper: Ultraheavy particle dark matter"

_SciPost Physics Core, doi:SciPost Phys. Core 6, 075 (2023)_

## Round 2 · Author Response

This is a resubmission following addressal of the referee's comments.

---

## Round 2 · List of Changes

Dear editor,

We thank the referee for their positive appraisal of our white paper
and recommendation to publish. We also appreciate their valuable
comments that we address below.
* * *
REFEREE

The detection sections focus on model-independent bounds on heavy dark
matter, which looks decoupled from the production mechanism. Some
discussion is given in the direct detection section but less in the
indirection part. The paper could benefit greatly from more
discussions on detecting ultra-heavy dark matter for several
mechanisms presented in Sec 2.

OUR RESPONSE

We agree. The DM decay signatures we had discussed are agnostic to the
exact cosmological production mechanism so long as particle DM (as
opposed to PBHs, etc.) is produced.

Accordingly, we have now added to the introductory paragraph of the
indirect detection section the following text.

"Ultraheavy DM models in this category may be generically produced in
the early universe by a number of mechanisms discussed above,
including freeze-out, freeze-in, gravitational production, and
involving phase transitions."
* * *
REFEREE

The signatures of heavy dark matter models could also be detected by
cosmological observations. The authors may consider adding some
discussions on this.

OUR RESPONSE

We appreciate the suggestion. To the end of the indirect detection
section we have added the following text.

"Finally, cosmological observations could also constrain ultraheavy
DM. CMB anisotropies would carry imprints of DM scattering with SM
matter, which may be exploited to probe a wide range of DM
masses~\cite{Dvorkin:2013cea,Gluscevic:2017ywp,Buen-Abad:2021mvc,Nguyen:2021cnb}.
Moreover, ultraheavy DM produced gravitationally is accompanied by
primordial non-Gaussianities that may be enhanced and observed in the
CMB power spectrum~\cite{Li:2019ves,Li:2020xwr}."
* * *
REFEREE

For gravitational particle production through inflation, the authors
wrote that the production is efficient when dark matter mass is
comparable to the inflationary Hubble scale. But I think the
production is efficient when the mass is much smaller than the Hubble
scale.

OUR RESPONSE

As we had noted, there may be exceptions, however gravitational
production as in WIMPzilla models is indeed generically most efficient
when the particle mass is comparable to the Hubble scale at the end of
inflation. This is stated, e.g., at the end of page 1 in
https://arxiv.org/abs/hep-ph/9802238.
* * *
With these changes we hope our manuscript will now be taken up for publication.

---

## Editorial Decision

published